# Critical Risks Method (CRM): A New Safety Allocation Approach for a Critical Infrastructure

**Gianpaolo Di Bona** [1],*[ID]**, Antonio Forcina** [2][ID]**, Domenico Falcone** [1]**and Luca Silvestri** [3][ID]

1   Department of Civil and Mechanical Engineering, University of Cassino and Southern Lazio,
    03043 Cassino FR, Italy; falcone@unicas.it
2   Department of Engineering, University of Naples "Parthenope", 80100 Naples, Italy;
    antonio.forcina@uniparthenope.it
3   Department of Engineering, University "Niccolò Cusano", 00166 Roma RM, Italy; luca.silvestri@unicusano.it
*   Correspondence: dibona@unicas.it; Tel.: +39-07762994331

**Abstract:** In the current research, a safety allocation technique named the Critical Risks Method (CRM) has been developed. Starting from a literature review, we analyzed the shortcomings of conventional methods. The outcomes show the primary two criticalities of the most important safety allocation approaches: (1) They are developed for series configuration, but not for parallel ones; (2) they ordinarily give only qualitative outputs, but not quantitative ones. Moreover, by applying the conventional methods, an increase in safety of the units to ensure the safety target leads to an increase of the production costs of the units. The proposed strategy can overcome the shortcomings of traditional techniques with a safety approach useful to series–parallel systems in order to obtain quantitative outputs in terms of failures in a year. The CRM considers six factors that are able to ensure its applicability to a great variety of critical infrastructures. In addition, CRM is described by a simply analytic definition. The CRM was applied to a critical infrastructure (Liquid Nitrogen Cooling Installation) in a nuclear plant designed with series–parallel units. By comparing the CRM outputs with databank safety values, the proposed method was validated.

**Keywords:** safety assessment; safety allocation; RAMS analysis; risk management; nuclear system

## 1. Introduction

Safety Instrument Systems (SISs) are units designed to ensure the safety of people and the environment. The international standard IEC 61508 [1] gives a safety approach to evaluate safety targets. This standard is conventional and subjective. Sector-specific standards are developed using IEC 61508; for example, IEC 61511 [2] for business analysis and IEC 62061 [3] for hardware frameworks. The standard gives a hazard analysis to evaluate the safety requirements of units. Starting from the safety target of the whole system, it is necessary to evaluate the safety value of the units. This approach is called safety allocation in IEC 61508. IEC 61508 and IEC 61511 recommend two methods for this approach: The Risk Graph method and the layers of protection analysis (LOPA). The Risk Graph method has been broadly discussed [4,5]. Many researches point out some shortcomings of the technique, in particular due to the subjective idea of the risk graph and risk matrix [6,7]. Baybutt (2007) recommends an improved hazard diagram technique to overcome these shortcomings. The LOPA technique was presented by the Center for Chemical Process Safety (CCPS) (1993) for industrial processes [8]. This methodology can be incorporated with a Hazard and Operability study (HAZOP). Numerous techniques have been developed [9,10]. All approaches give qualitative outputs [11,12]. The European Space Agency (ESA) has created a quantitative approach: The Sphynx Method [13]. The ESA's approach has been structured to allocate safety targets to aerospace systems. The examined

techniques share a shortcoming in their scientific formulations: They are developed for units with series configurations, but not for series–parallel configurations. Furthermore, only the Sphynx Method provides quantitative results. In order to overcome these criticalities, a new safety allocation approach has been proposed and validated: The Critical Risks Method (CRM). The new technique was applied on a toroidal machine, which is important in completing research on plasma material and controlled atomic fusion. A nuclear plant was structured using series–parallel configurations in order to ensure safety. This paper is organized as follows: Section 2 introduces the Nuclear System, Section 3 analyzes the state of the art of safety allocation techniques, in Section 4, the framework of the CRM is described, and, finally, in Section 5, a case study is presented. Section 6 summarizes the conclusions of the research and the future developments.

## 2. Nuclear System

Nuclear fusion [14] is a strongly energetic reaction: Two "light" particles (with low nuclear number), for example, hydrogen or its isotopes, deuterium and tritium, are fused to deliver heavier atoms, like helium.

The nuclei of hydrogen (H), deuterium (D), and tritium (T) contain one proton and alternate quantities of neutrons; one for the nucleus of deuterium, two for tritium. In each of the three cases, the particle, electrically neutral, has an electron orbiting around the nucleus, compensating the single proton charge. Regularly, a nucleus of deuterium and one of tritium are combined to deliver a nucleus of helium (alpha particle) and a neutron (Figure 1).

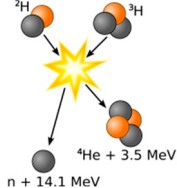

**Figure 1.** Fusion reaction.

At the end of the reaction, the total mass is lower than that of the interacting elements. The difference, called defect of mass, transforms into energy, according to the Einstein's notable law:

$$E = m \times c^2. \tag{1}$$

In order to obtain energy production through controlled nuclear reactions, it is important to heat the plasma of deuterium–tritium up to extremely high temperatures (around $10^8$ °C), keeping the hot plasma confined in a magnetic field, to force particles to follow spiral trajectories. In magnetic confinement, hot plasma is enclosed inside a vacuum chamber. In the present research, we analyzed a toroidal machine (Figure 2) [15].

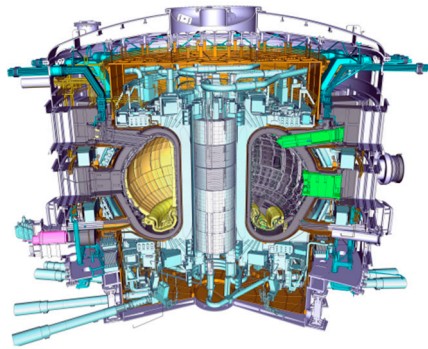

**Figure 2.** Toroidal machine (source: ENEA Frascati—Italy).

In order to cool the vacuum chamber and coils, a closed circuit of liquid nitrogen was designed with the following units (Figure 3):

- Three buffer tanks of fluid nitrogen with an all-out limit of 90,000 L and pressure of 2.5 bar;
- Two cryogenic pumps lubricated by a similar fluid nitrogen;
- Two evaporators;
- Tanks, valves, and common extras.

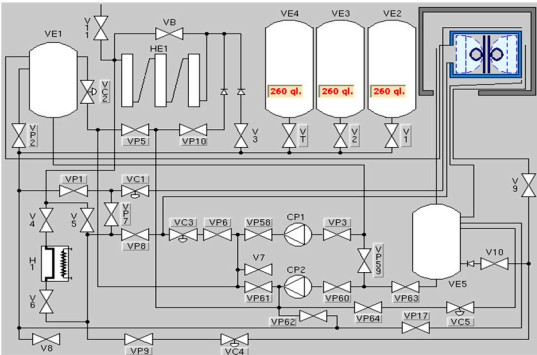

**Figure 3.** Cooling system.

The cryostat is the main unit of the system. In order to cool the main components, the nitrogen pipes arrive at the cryostat. The toroidal framework is allocated inside the cryostat, where the pressure is higher than outside (20 mm $H_2O$) in order to avoid the entry of atmosphere (working temperature of −190 °C) [16].

## 3. Literature Review: State of the Art of Safety Allocation Methods

In the present section, the conventional methodologies of safety allocation are analyzed.

Let S*(t) (events/time) be the safety target of a series system. Let Si*(t) (events/time) be the safety allocation for unit i [17,18]:

$$S_i^*(t) = S^*(t) \cdot w_i\%  \tag{2}$$

The allocation is an iterative procedure in order to define $w_i\%$. It begins from the design phase, when little information about the units is available. In this stage, it is smarter to consider units in series. The initial step of the safety allocation process is to allocate the safety target to all units. IEC 61508 does not give any conventional techniques to allocate safety targets.

IEC 61508 recommends some methodologies for this purpose:

- The "As Low As Reasonably Practicable" method (ALARP);
- The Risk Graph method;
- Layers of Protection Analysis (LOPA);
- Hazardous event severity matrix.

Another method, not suggested by IEC 61508, is the Sphynx Method. This approach was developed by the ESA.

### 3.1. ALARP Method

The ALARP method is described by the risk triangle: (a) Unacceptable risk (red color) on the top, (b) tolerable risk in the middle (yellow color), and (c) acceptable risk (green color) at the bottom (Figure 4). The risk degree decreases from high to low through mitigations or measures. Safety allocated above the red level is intolerable and risk reduction is necessary. Between the red level and the green level, the risk is only tolerable if it is ALARP, which means that all reasonably practicable risk reduction

measures have been identified and implemented. The reduction of safety cost (money, time, or effort) is greater than the reduction of the safety target. In other words, ALARP is simply a balancing of risk reduction and the cost to achieve it.

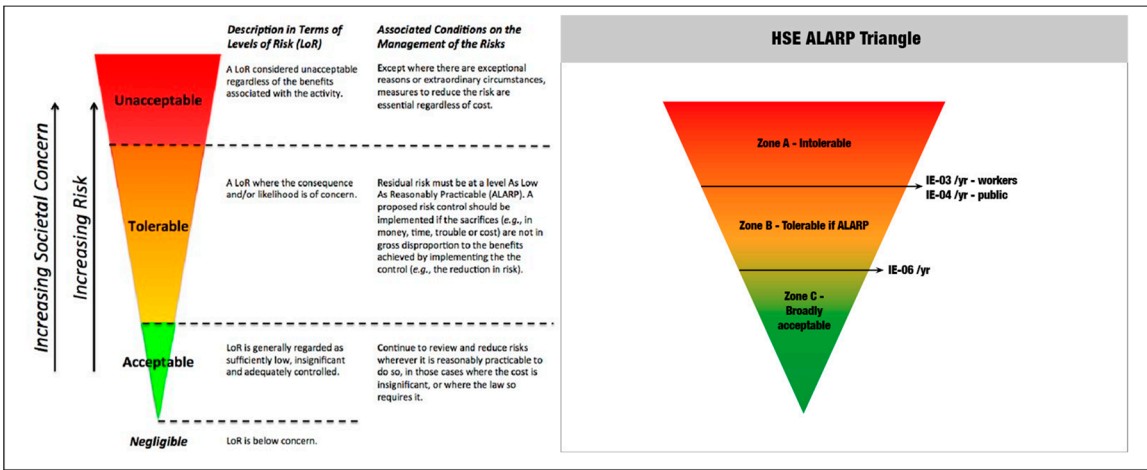

**Figure 4.** "As Low As Reasonably Practicable" (ALARP) method.

The risk management has to demonstrate that a risk is ALARP. In order to implement risk reduction measures, it is important to determine the correct approach to assess whether it is ALARP or not. According to the ALARP method, the appropriate techniques could be: (a) engineering judgement, (b) qualitative risk assessment, or c) semi-quantitative risk assessment.

There are some clear strengths with this approach:

- It is easy to understand and apply.
- However, there are numerous weaknesses and limitations:
- It is qualitative methodology;
- It is very difficult to define an objective $w_i$ for every unit.

The qualitative methodologies do not allow an accurate evaluation of safety values. They express only a judgment influenced by the experts. Quantitative methods, on the other hand, permit an estimate of the safety value expressed as faults per year.

*3.2. The Risk Graph Method*

The Risk Graph technique allows the valuation of the safety target according to the hazard factors of units. The technique is useful for safety allocation of mechanical equipment (IEC 62061, 2005, Annex A) or industrial systems (IEC 61511, 2003, Part 3), and should be used in the chemical process (Salis, 2011). The approach is useful for qualitative and quantitative risk assessment. A wide variety of factors that define the nature of the units are used. According to IEC 61508, the necessities for the preceding parameters have to enable a significant ranking of the danger, and additionally have to include the key elements for danger evaluation. The standard offers a simplified process and an established scheme, introduced in Figure 5. This normal instance uses four factors to define units (IEC 61508, 2010, Annex E, Section 5).

Safety requirements range from unrequired through the Safety Integrity Levels (SILs) 1–4. Safety, environmental, and economic impact are pursued by the Risk Graph method. The safety percentile weight is:

$$w_i\% = \frac{SIL_i}{\sum_{i=1}^{n} SIL_i}. \tag{3}$$

There are also some clear strengths with this approach:

- It can be conducted both qualitatively and quantitatively;
- It is easy to understand and apply.
- However, there is a great limitation:
- It is only suitable for series configuration.

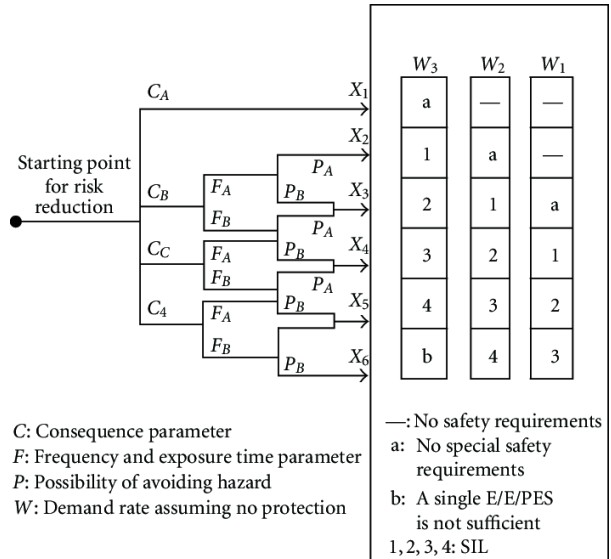

**Figure 5.** Risk graph—general scheme (reproduced from IEC 61508).

### 3.3. LOPA Method

The LOPA method is a semi-quantitative risk assessment technique introduced by the Center for Chemical Process Safety in 1993 (CCPS, 1993). The motivation behind LOPA is to decide if there are adequate safety levels against explicit accident situations (CCPS, 2001). A safety layer in LOPA is equivalent to a safety unit. In addition, CCPS (2001) introduced the idea of independent safety layers. The necessities for an independent protection layer (IPL) are referred to in IEC 61511 (2003, Part 3).

The LOPA method generally follows an HAZOP analysis. An LOPA event tree (Figure 6) can represent the different accident situations for a critical system. In this example, the specific initiating event can result in one out of four end events.

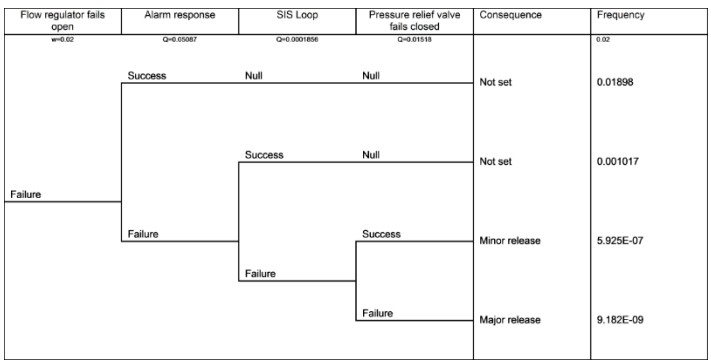

**Figure 6.** Layers of Protection Analysis (LOPA) event tree.

According to this method, the safety percentile weight is:

$$w_i\% = \frac{SIL_i}{\sum_{i=1}^{n} SIL_i}.$$

(4)

There are some clear strengths with this approach:

- It can be conducted both qualitatively and quantitatively;
- It is incorporated into HAZOP analysis.

However, there are also weaknesses and limitations:

- It is only suitable for low-demand systems;
- It is only suitable for series configuration.

### 3.4. Hazardous Event Severity Matrix

Starting from Failure Mode and Effect Analysis (FMEA), the Risk Priority Number (RPN) has been used by recent researches to consider the failure effect in reliability. Let unit i have $N_j$ failure modes with severity ranking $S_{ij}$, occurrence rating $O_{ij}$, and detection ranking $D_{ij}$. The three factors are evaluated by an ordinal scale from 1 to 10. The RPN of failure mode j in unit i is given by the following equation:

$$RPN = S_{ij} \times O_{ij} \times D_{ij.} \tag{5}$$

The lack of objectivity and the difficulty of risk effect comparison are the shortcomings of this approach [19]. It is a semi-quantitative method. The O and S values are determined on a quantitative and semantic scale defined by various international standards, such as IEC 60812 (2006) [20] and ISO 31010 (2010) [21].

The safety percentile weight is:

$$w_i = \frac{\omega_i}{\sum\limits_{i=1}^{k} \omega_i} \text{ where } \omega_i = \frac{C_i}{\sum\limits_{i=1}^{k} C_i} \text{ and } C_i = \frac{1}{N_i} \sum\limits_{j=1}^{N_i} (S_{ij} \times O_{ij} \times D_{ij}) \tag{6}$$

There are some clear strengths with this approach [22]:

- It can be conducted both qualitatively and quantitatively;
- It is very simple.

However, there is a main limitation:

- It is only suitable for series configuration.

**Sphynx Method**

The Sphynx approach was structured to allocate safety targets to the ESA's aerospace prototypes [23]. The Sphynx method is based on "Allocation Factors" $AF_i$ for unit i. The formulation is the following:

$$AF_i = D_e + [(D_t + F) \cdot C_f] \tag{7}$$

where:

$D_e$ = Environmental risks;
$D_t$ = Technological risks;
F = Number of catastrophic/critical/marginal/minor functions of the system;
C = Complexity index.

The complexity index value ranges between 0 and 1, and is obtained by normalizing the Complexity Factor (Table 1) ($C_f$):

$$C_{fi} = [\text{Technology} \cdot \text{Dimension} \cdot \text{Material} \cdot \text{Time}] \tag{8}$$

**Table 1.** Complexity Factor.

| Factor | Min | Max |
|--------|-----|-----|
| Technology | 1 | 10 |
| Dimension | 1 | 10 |
| Material | 1 | 10 |
| Time | 1 | 10 |

The safety percentile weight is:

$$w_i\% = \frac{AF_i}{\sum_{i=1}^{n} AF_i}. \tag{9}$$

There are some clear strengths with this approach:

- It is quantitative methodology;
- It is suitable for complex systems where high safety standards are required.

However, there are some weaknesses and limitations:

- The sum of the number of functions with technological risks has no scientific reason;
- It is only suitable for series configuration.

The review of the literature techniques points out that there are significant difficulties in conducting an objective safety allocation. All of the suggested methods have their strengths and weakness, which have been illustrated in this section.

The Risk Graph method has a few significant criticalities. Baybutt (2014) suggested that the approach has a narrow application area.

The LOPA methodology is easier to understand and is integrated with HAZOP. The technique takes into consideration various parameters, e.g., safety, failures, environmental impact, multiple units, etc. The biggest issue with the LOPA model is that it cannot be used on SIL 3 or SIL 4 systems.

The hazardous event severity matrix approach is probably sufficient to achieve a tolerable risk, but it is questioned if this method will survive, as it leads to overly conservative safety requirements.

The Sphynx approach shows some limitations. In particular, the technique was developed for an aerospace environment. In this situation, environmental hazards are more important than technological ones ($D_e$ are considered integrally, whereas $D_t$ are considered partially).

A more flexible formulation could consider that:

- Environmental and technological hazards should have the same importance;
- It is necessary to value the real influence of each hazard on the considered unit.

The analysis of the Sphynx method points out a shortcoming concerning the F factor, since the sum of F with $D_t$ is not established on scientific reason.

Starting from the above considerations, we proposed a new safety allocation technique in order to solve the limitations of conventional approaches.

The above analysis has suggested some guidelines to develop the new method. We have applied the most suitable approach for the nuclear system. In particular, we applied the RPN and Sphynx methods because:

- Thermonuclear systems are in the production phase—many factors are known (system criticality, technology, mission time, etc.);
- Thermonuclear systems have a complexity similar to that of aerospace;
- Thermonuclear system detection is an important parameter for safety allocation.

## 4. Critical Risks Method

The correct environmental condition for the toroidal system is the mission of the cooling unit in order to confine plasma in magnetic fields. Appling a Safety Block Diagram (SBD), the whole system has been de-structured into functional units in series configuration (Table 2):

**Table 2.** Units of the cooling system.

| | |
|---|---|
| Unit 1. Manual valves. | Unit 11. Cryostat. |
| Unit 2. Safety valves. | Unit 12. Liquid nitrogen tanks. |
| Unit 3. Restraint valves. | Unit 13. Separation tank. |
| Unit 4. On-off valves. | Unit 14. Collection tank. |
| Unit 5. Solenoid valves. | Unit 15. Main evaporators. |
| Unit 6. Breaking discs. | Unit 16. Secondary evaporators. |
| Unit 7. Pressure valves. | Unit 17. Heater. |
| Unit 8. Self-regulation valves. | Unit 18. Cryogenic pumps. |
| Unit 9. Pressure-regulation valves. | Unit 19. Compressed air system. |
| Unit 10. Level valves. | Unit 20. Measure modules. |

Then, Top Events were developed through a Preliminary Hazard Analysis (PHA) (Table 3).

**Table 3.** Preliminary Hazard Analysis (MIL-STD1629A).

| TOP EVENT | Minor | Marginal | Critical | Catastrophic |
|---|---|---|---|---|
| Frequent | | | | |
| Probable | Cooling Cycle Interruption | | | |
| Occasional | | | Low Pressure in Cryostat | |
| Rare | | | | Damage in Cryostat |
| Improbable | | | | |

According to an expert judgment, a safety target was evaluated in terms of faults per year (Table 4).

**Table 4.** Safety target values (life cycle = 25 years).

| TOP Event | Accepted Faults/Mission | Safety Target (Faults/Year) |
|---|---|---|
| T.E.1: Catastrophic | 1/1000 | 0.002 |
| T.E.2: Critical | 1/500 | 0.004 |
| T.E.3: Minor | 1/250 | 0.008 |

Starting from an Functional-FMECA analysis, it was possible to estimate the allocation indexes for the RPN (Table 5) and Sphynx methods (Table 6), only in series configuration. The results show how the safety target influenced the allocated values [24] during the working and maintenance phases [25].

**Table 5.** Risk Priority Number (RPN) method.

| Unit | | | | | | T.E. 1 | T.E. 2 | T.E. 3 |
|---|---|---|---|---|---|---|---|---|
| | **S** | **O** | **D** | **RPN** | **W%** | **0.002** | **0.004** | **0.008** |
| 1. Manual valves | 5 | 5 | 6 | 150 | 2.44% | $4.89 \times 10^{-5}$ | $1.96 \times 10^{-7}$ | $1.56 \times 10^{-9}$ |
| 2. Safety valves | 10 | 5 | 6 | 300 | 4.89% | $9.78 \times 10^{-5}$ | $3.91 \times 10^{-7}$ | $3.13 \times 10^{-9}$ |
| 3. Restraint valves | 6 | 5 | 5 | 150 | 2.44% | $4.89 \times 10^{-5}$ | $1.96 \times 10^{-7}$ | $1.56 \times 10^{-9}$ |
| 4. On–off valves | 8 | 9 | 9 | 648 | 10.56% | $2.11 \times 10^{-4}$ | $8.45 \times 10^{-7}$ | $6.76 \times 10^{-9}$ |
| 5. Solenoid valves | 9 | 9 | 5 | 405 | 6.60% | $1.32 \times 10^{-4}$ | $5.28 \times 10^{-7}$ | $4.22 \times 10^{-9}$ |
| 6. Breaking discs | 2 | 3 | 5 | 30 | 0.49% | $9.78 \times 10^{-6}$ | $3.91 \times 10^{-8}$ | $3.13 \times 10^{-10}$ |
| 7. Valves at static | 5 | 5 | 8 | 200 | 3.26% | $6.5 \times 10^{-5}$ | $2.61 \times 10^{-7}$ | $2.09 \times 10^{-9}$ |
| 8. Self-regulation valves | 6 | 6 | 7 | 252 | 4.11% | $8.22 \times 10^{-5}$ | $3.29 \times 10^{-7}$ | $2.63 \times 10^{-9}$ |
| 9. Pressure-regulation valves | 8 | 9 | 6 | 432 | 7.04% | $1.41 \times 10^{-4}$ | $5.63 \times 10^{-7}$ | $4.51 \times 10^{-9}$ |
| 10. Level valves | 4 | 6 | 8 | 192 | 3.13% | $6.26 \times 10^{-5}$ | $2.50 \times 10^{-7}$ | $2.00 \times 10^{-9}$ |
| 11. Cryostat | 5 | 1 | 9 | 45 | 0.73% | $1.47 \times 10^{-5}$ | $5.87 \times 10^{-8}$ | $4.69 \times 10^{-10}$ |
| 12. Liquid nitrogen tanks | 6 | 5 | 6 | 180 | 2.93% | $5.87 \times 10^{-5}$ | $2.35 \times 10^{-7}$ | $1.88 \times 10^{-9}$ |
| 13. Separation tank | 8 | 9 | 8 | 576 | 9.39% | $1.88 \times 10^{-4}$ | $7.51 \times 10^{-7}$ | $6.01 \times 10^{-9}$ |
| 14. Collection tank | 9 | 7 | 7 | 441 | 7.19% | $1.44 \times 10^{-4}$ | $5.75 \times 10^{-7}$ | $4.60 \times 10^{-9}$ |
| 15. Main evaporators | 4 | 5 | 8 | 160 | 2.61% | $5.22 \times 10^{-5}$ | $2.09 \times 10^{-7}$ | $1.67 \times 10^{-9}$ |
| 16. Secondary evaporators | 5 | 4 | 6 | 120 | 1.96% | $3.91 \times 10^{-5}$ | $1.56 \times 10^{-7}$ | $1.25 \times 10^{-9}$ |
| 17. Heater | 6 | 8 | 9 | 432 | 7.04% | $1.41 \times 10^{-4}$ | $5.63 \times 10^{-7}$ | $4.51 \times 10^{-9}$ |
| 18. Cryogenic pumps | 9 | 6 | 9 | 486 | 7.92% | $1.58 \times 10^{-4}$ | $6.34 \times 10^{-7}$ | $5.07 \times 10^{-9}$ |
| 19. Compressed air system | 9 | 4 | 8 | 288 | 4.69% | $9.39 \times 10^{-5}$ | $3.76 \times 10^{-7}$ | $3.00 \times 10^{-9}$ |
| 20. Measure modules | 9 | 9 | 8 | 648 | 10.56% | $2.11 \times 10^{-4}$ | $8.45 \times 10^{-7}$ | $6.76 \times 10^{-9}$ |

**Table 6.** Sphynx method.

| Unit | | $D_e$ | $D_t$ | F | C | AF | W% | T.E. 1 0.002 | T.E. 2 0.004 | T.E. 3 0.008 |
|------|---|-------|-------|---|---|----|----|----|----|----|
| 1. | Manual valves | 2 | 1 | 0.02 | 95 | 98.9 | 0.54% | $1.09 \times 10^{-5}$ | $4.34 \times 10^{-8}$ | $3.48 \times 10^{-10}$ |
| 2. | Safety valves | 3 | 2 | 0.1 | 126 | 267.6 | 1.47% | $2.94 \times 10^{-5}$ | $1.18 \times 10^{-7}$ | $9.40 \times 10^{-10}$ |
| 3. | Restraint valves | 5 | 3 | 0.04 | 142 | 436.68 | 2.40% | $4.80 \times 10^{-5}$ | $1.92 \times 10^{-7}$ | $1.53 \times 10^{-9}$ |
| 4. | On–off valves | 5 | 1 | 0.02 | 235 | 244.7 | 1.34% | $2.69 \times 10^{-5}$ | $1.07 \times 10^{-7}$ | $8.60 \times 10^{-10}$ |
| 5. | Solenoid valves | 3 | 4 | 0.05 | 530 | 2149.5 | 11.80% | $2.36 \times 10^{-4}$ | $9.44 \times 10^{-7}$ | $7.55 \times 10^{-9}$ |
| 6. | Breaking discs | 1 | 3 | 0.05 | 159 | 485.95 | 2.67% | $5.34 \times 10^{-5}$ | $2.13 \times 10^{-7}$ | $1.71 \times 10^{-9}$ |
| 7. | Valves at static | 2 | 1 | 0.05 | 235 | 248.75 | 1.37% | $2.73 \times 10^{-5}$ | $1.09 \times 10^{-7}$ | $8.74 \times 10^{-10}$ |
| 8. | Self-regulation valves | 3 | 3 | 0.05 | 256 | 783.8 | 4.30% | $8.61 \times 10^{-5}$ | $3.44 \times 10^{-7}$ | $2.75 \times 10^{-9}$ |
| 9. | Pressure-regulation valves | 4 | 2 | 0.05 | 568 | 1168.4 | 6.41% | $1.28 \times 10^{-4}$ | $5.13 \times 10^{-7}$ | $4.11 \times 10^{-9}$ |
| 10. | Level valves | 4 | 2 | 0.05 | 452 | 930.6 | 5.11% | $1.02 \times 10^{-4}$ | $4.09 \times 10^{-7}$ | $3.27 \times 10^{-9}$ |
| 11. | Cryostat | 4 | 2 | 0.06 | 148 | 308.88 | 1.70% | $3.39 \times 10^{-5}$ | $1.36 \times 10^{-7}$ | $1.09 \times 10^{-9}$ |
| 12. | Liquid nitrogen tanks | 3 | 4 | 0.04 | 239 | 968.56 | 5.32% | $1.06 \times 10^{-4}$ | $4.25 \times 10^{-7}$ | $3.40 \times 10^{-9}$ |
| 13. | Separation tank | 2 | 3 | 0.1 | 215 | 668.5 | 3.67% | $7.34 \times 10^{-5}$ | $2.94 \times 10^{-7}$ | $2.35 \times 10^{-9}$ |
| 14. | Collection tank | 3 | 4 | 0.07 | 786 | 3202.02 | 17.58% | $3.52 \times 10^{-4}$ | $1.41 \times 10^{-6}$ | $1.13 \times 10^{-8}$ |
| 15. | Main evaporators | 4 | 2 | 0.03 | 125 | 257.75 | 1.42% | $2.83 \times 10^{-5}$ | $1.13 \times 10^{-7}$ | $9.06 \times 10^{-10}$ |
| 16. | Secondary evaporators | 2 | 3 | 0.05 | 369 | 1127.45 | 6.19% | $1.24 \times 10^{-4}$ | $4.95 \times 10^{-7}$ | $3.96 \times 10^{-9}$ |
| 17. | Heater | 5 | 1 | 0.04 | 357 | 376.28 | 2.07% | $4.13 \times 10^{-5}$ | $1.65 \times 10^{-7}$ | $1.32 \times 10^{-9}$ |
| 18. | Cryogenic pumps | 3 | 2 | 0.04 | 159 | 327.36 | 1.80% | $3.59 \times 10^{-5}$ | $1.44 \times 10^{-7}$ | $1.15 \times 10^{-9}$ |
| 19. | Compressed air system | 2 | 3 | 0.08 | 754 | 2324.32 | 12.76% | $2.55 \times 10^{-4}$ | $1.02 \times 10^{-6}$ | $8.17 \times 10^{-9}$ |
| 20. | Measure modules | 1 | 4 | 0.01 | 458 | 1837.58 | 10.09% | $2.02 \times 10^{-4}$ | $8.07 \times 10^{-7}$ | $6.46 \times 10^{-9}$ |

The analysis of the RPN results shows that (1) there are high values of allocated safety (series configuration), and (2) the standard deviation is high—there is a big difference between safety values.

The analysis of the Sphynx results shows that (1) there are high values of allocated safety (series configuration), (2) there are some low values, and (3) the standard deviation is low.

There is not any reference to a potential "buffer effect" (parallel configuration); in fact, in the Safety Block Diagram (SBD), there are only series configurations [26]. Table 7 summarizes these comparisons.

**Table 7.** Comparison of methods.

| Method | Fitting to Nuclear System | Not Fitting to Nuclear System |
|---|---|---|
| RPN | Index factors oriented safety allocation. | Functional factors oriented safety allocation. The buffer effect is not evaluated. |
| Sphynx | Index factors oriented safety allocation. Units and Top Events are linked. The allocation factors are based on operating and environmental conditions of units. | No information on operating cycles. The buffer effect is not evaluated. Functional importance is the same for all units. |

Starting from the above outputs, a new safety approach was proposed. The guidelines to develop an allocation technique are:

√　Generality;
√　Standardization of inputs;
√　Economy;
√　Realistic and achievable requirements.

The proposed allocation approach, named the "Critical Risks Method", was developed for the toroidal machine [27], but it can also be useful for any critical infrastructure (series and parallel configuration).

The first stage was the examination of critical units according to expert judgment. In order to restrain the analysis to low number of components, a critical unit ranking was developed [28]. The CRM is structured in the following steps:

**Step 1**: Definition of the system and units;
**Step 2**: Construction of a Safety Block Diagram (SBD);
**Step 3**: Analysis of a Preliminary Hazard Analysis (PHA) of the Top Events;
**Step 4**: F-FMECA analysis to point out catastrophic/critical/marginal/minor functions of unit i;
**Step 5**: Calculation of $A_1$, $A_2$, $A_3$, $A_4$, $A_5$, and $A_6$ as factors for every unit, where:

*Criticality Factor ($A_1$):* It allows evaluation of the consequences on a Top Event caused by a total or partial unit failure. The factor will assign higher safety to less critical systems. The index can vary between 0 (n = ∞) for a low criticality of the unit and 1 (n = 1) for highly critical one. The $A_1$ factor is evaluated through the following equation:

$$A_1 = \frac{1}{n},\qquad(10)$$

where "n" is the number of "buffer elements" (parallel configuration) that can oppose a risk implementation. The factor permits the assignment of a low safety value to a parallel configuration (n > 1).

*Environmental Risk Factor ($A_2$):* It allows the evaluation of the stress level caused by environmental factors for a single unit. The factors will assign higher safety to the most stressed unit.

$$A_2 = 1 - \frac{1}{f_i},\qquad(11)$$

where the $f_i$ value ranges between 1 and 100:

- f = 1 means a little influence of environmental conditions on unit i;
- f = 100 means a great influence of environmental conditions on unit i.

It could be difficult to estimate the f value in the pre-design phase. However, a simple evaluation will be possible in developed critical infrastructures, thanks to professional judgment supports and by comparison to similar structures.

*Technological Risk Factor (A₃):* It allows the evaluation of the stress level caused by technological factors for a single unit. The factors will assign higher safety to the most technologically advanced unit.

$$A_3 = 1 - \frac{1}{g_i}, \tag{12}$$

where the $g_i$ value ranges between 1 and 100:

- g = 1 means a little influence of technological conditions on unit i;
- g = 100 means a great influence of technological conditions on unit i.

It could be difficult to estimate the f value in the pre-design phase. However, a simple evaluation it will be possible in developed critical infrastructures, thanks to professional judgment supports and by comparison to similar structures. This represents the technological level of a single unit.

*Functionality Factor (A₄):* The factor evaluates the functionality of the units in terms of structure, assembly, and interactions. It permits one to discriminate the system unit complexity, linked to the number of functions.

$$A_4 = \frac{H_i}{K_i} \tag{13}$$

Event factor K is

$$K_i = \frac{n.catastrofic/critical/marginal/minor\_subsystem_i\_fuctions}{n.\_subsystem_i\_fuctions}, \tag{14}$$

where the K numerator is the number of functions that may cause a catastrophic/critical/marginal/minor event.

In addition, functionality factor H is:

$$H_i = \frac{n.\_subsystem_i\_fuctions}{n\_system\_fuctions}, \tag{15}$$

where the H denominator is the ratio between the number of unit functions and the number of system functions. The H factor discriminates the system units' complexity, linked to the number of functions.

Functionality factors assign a high safety target to critical units (high H factor, low K factor), as opposed to the Sphynx method, which assigns a low safety target to a unit with many critical functions.

*Complexity Factor (A₅):* See Sphynx method (Equation (7))

**Step 6:** Calculation of the Allocated Factor of unit i:

$$AF_i = (A_{i1}A_{i2}A_{i3}A_{i4}A_{i5}) \; i = 1 \ldots k. \tag{16}$$

Calculation of the Allocated Safety Weight of unit i:

$$w_i = \frac{AF}{\sum\limits_{j=1}^{n} AF_j} = \frac{(A_{i1}A_{i2}A_{i3}A_{i4}A_{i5})}{\sum\limits_{i=1}^{k} (A_{i1}A_{i2}A_{i3}A_{i4}A_{i5})} \; i = 1 \ldots k, \tag{17}$$

where $w_i$ is the global weight of the *i*-th unit. After the evaluation of $w_i$, it is possible to allocate the safety target using Equation (2):

**Step 7:** Analysis of results.

## 5. Application of the CRM

The proposed approach was applied to the thermonuclear system described in Section 2. According to Preliminary Hazard Analysis (PHA) (Table 3), the proposed approach was applied for each Top Event.

**Step 1:** Analyzed in Section 4.

**Step 2:** The reality of the Safety Block Diagram of the cooling system is a series–parallel configuration. In reality, not all of the units shown in Table 2 are related to every Top Event. The SBD for each of the three Top Events was modified starting from functional and FMECA tables. The CRM permits the evaluation of the subgroups of units, influencing Top Events with their "buffer units" (parallel configuration). Figure 7 describes the safety block diagram for the second Top Event (low pressure in the cryostat) [29].

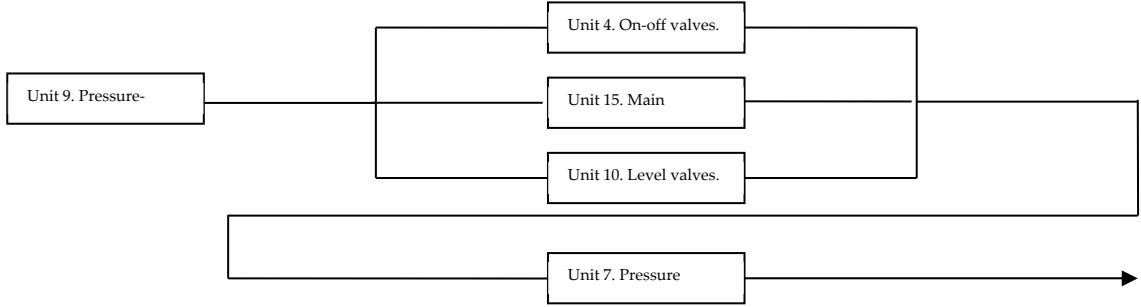

**Figure 7.** Safety Block Diagram for the critical event.

The SBD shows that a constant pressure value depends on the cycle of pressurization, but also on the presence of liquid nitrogen in the collection tank. Some nitrogen present in the tank evaporates, contributing to maintaining a fixed level of pressure in the cryostat [30]:

**Step 3:** Analyzed in Section 4.

**Step 4:** Functional analysis (Table 8) and FMECA (Table 9) analysis were implemented to point out catastrophic/critical/marginal/minor functions of Units 9, 4, 15, 10, and 7.

**Table 8.** Functional analysis of Unit 9.

| Unit 9—Pressure-Regulation Valves | | | |
|---|---|---|---|
| **Functions** | **Mode** | **Note** | **Linked Units** |
| Control valve opening VC4 according to the pressure of cryostat during the PLC cycle. | The opening is partial. Nominal pressure of c.a. 20 mm. | Incorrect functioning of the valve and its equipment (pressure sensors) could increase pressure in the cryostat. | Unit 11 |
| Control valve opening VC1 according to the cooling gradient magnets. | The opening is partial. | If the valve is opened excessively, a considerable amount of nitrogen is discharged into the cryostat and into the reservoir. | Unit 11 |
| Setting pressure of the CP1 pump by VC2. | The opening is partial. | Nothing | Nothing |
| Setting pressure in the copper cooling circuit using VC3. | The opening is partial. Nominal pressure of 1.5 bar. | After the shot, the fluid heats up by increasing the volume. The valve prevents excessive pressure increase due to fluid mass input. | Unit 2 |

**Table 9.** FMECA Analysis of Unit 9.

| Unit 9—Pressure-Regulation Valves | | | | | |
|---|---|---|---|---|---|
| **Functions** | **Failure Mode** | **Causes** | **Effects** | **Corrective Actions** | **Note** |
| Control valve opening VC4 according to the pressure of cryostat during the PLC cycle. | Failure—Power Loss | Wear; Electrical supply interruption of the compressors and failure of start-up of auxiliary generators. | The nitrogen level increases | 1.1: Maintenance; 1.2: Restore compressed air supply | Nothing |
| Control valve opening VC1 according to the cooling gradient magnets. | Failure | Wear | The nitrogen level increases | 2.1: Maintenance | Dispersion of nitrogen |
| Setting pressure of the CP1 pump using VC2. | Failure—Power Loss | Wear; Electrical supply interruption of the compressors and failure of start-up of auxiliary generators. | The nitrogen level increases | 3.1: Maintenance | Nothing |
| Setting pressure in the copper cooling circuit using VC3. | Failure | Wear | The nitrogen level increases | 4.1: Maintenance | Damage of the cryostat |

**Step 5**: According to Equations (7) and (10)–(13), the allocations of the indexes were evaluated (Table 10).

**Table 10.** Allocations of the units' indexes.

| Elements | $A_1$ | $A_2$ | $A_3$ | $A_4$ | $A_5$ |
|---|---|---|---|---|---|
| Unit 9 | 1 | 0.98 | 0.90 | 50.00 | 235.00 |
| Unit 4 | 0.33 | 0.95 | 0.90 | 20.00 | 235.00 |
| Unit 15 | 0.33 | 0.98 | 0.95 | 20.00 | 568.00 |
| Unit 10 | 0.33 | 0.98 | 0.95 | 20.00 | 452.00 |
| Unit 7 | 1 | 0.98 | 0.95 | 33.00 | 125.00 |

**Step 6:** According to Equations (16) and (17), the safety allocation weights were evaluated. Then, according to Equation (2), the safety allocations were evaluated for single units (Table 11) for Top Event 1 (Catastrophic):

**Table 11.** Safety allocation.

| Elements | AF | $w_i$ | S(t) |
|---|---|---|---|
| Unit 9 | 10,363.50 | 47.66% | $9.53 \times 10^{-04}$ |
| Unit 4 | 1326.11 | 6.10% | $1.22 \times 10^{-04}$ |
| Unit 15 | 3472.33 | 15.97% | $3.19 \times 10^{-04}$ |
| Unit 10 | 2763.19 | 12.71% | $2.54 \times 10^{-04}$ |
| Unit 7 | 3820.78 | 17.57% | $3.51 \times 10^{-04}$ |

**Step 7:** The CRM's outputs show two problems related to Units 7 and 9 (level regulation valve and pressure regulation valve). In order to reduce the risk of the above units, the new approach suggests to

fill the collection tank through Unit 7. The result is an increase of the pressurization of the cryostat. A similar critical state is highlighted in the cycle of pressurization. In fact, a failure of Unit 9 could close the access to the gaseous nitrogen cryostat. In the same cycle, the Unit 10 shows less importance. The reason is that Unit 10 works in less stressful operating conditions because the number of opening and closing cycles is reduced.

In order to verify the CRM, safety targets were compared to allocated safety values [31]. Subsequently, the results obtained were compared, in terms of Mean Absolute Deviation (MAD) and negative technological errors (Table 12), with the results obtained in Section 4 (Tables 6 and 7).

In particular, the negative technological error is defined according to Equation (18).

$$\varepsilon_{\text{technological}_i} = [w_i \cdot S^*(t) - S(t)_{databanks}] \tag{18}$$

If $S(t)_{allocated} < S(t)_{databanks}$, we obtain a negative technological error. The $\varepsilon_{\text{technological}}$ values highlight the criticalities of the allocation technique, particularly the sum $\sum(-\acute{\varepsilon}_{\text{technological}})$.

The sum $\sum(-\acute{\varepsilon}_{\text{technological}})$ for CRM is the minimum ($\sum(-\acute{\varepsilon}) = -4.51 \times 10^{-3}$ (faults/year)) in relation with $\sum(-\acute{\varepsilon}_{\text{technological}})$ for RPN and $\sum(-\acute{\varepsilon}_{\text{technological}})$ for Sphynx.

The MAD for CRM is the minimum ($MAD_{CRM} = 5.16 \times 10^{-4}$) in relation with MAD for RPN and MAD for Sphynx.

The results obtained with CRM can be summarized as follows:

- Reduction of $S(t)_{allocat}$ for Units 4, 15 and 10 (parallel configuration); the average value is 28.61%. This means a good alignment with respect to the databank and a substantial savings in the choice of less-performing units;
- Reduction of MAD of about 32.93%;
- Reduction of $\sum(-\acute{\varepsilon}_{\text{technological}})$ of about 38.69%.

The results highlight that CRM assigns smaller values allocated to the components compared to databanks (expect Unit 15). These values ensure a "safety" condition for the nuclear units.

It is possible to notice that:

- The allocated safety values are comparable to the supplied safety ones;
- The units' performance and hierarchy are respected (Figure 8).

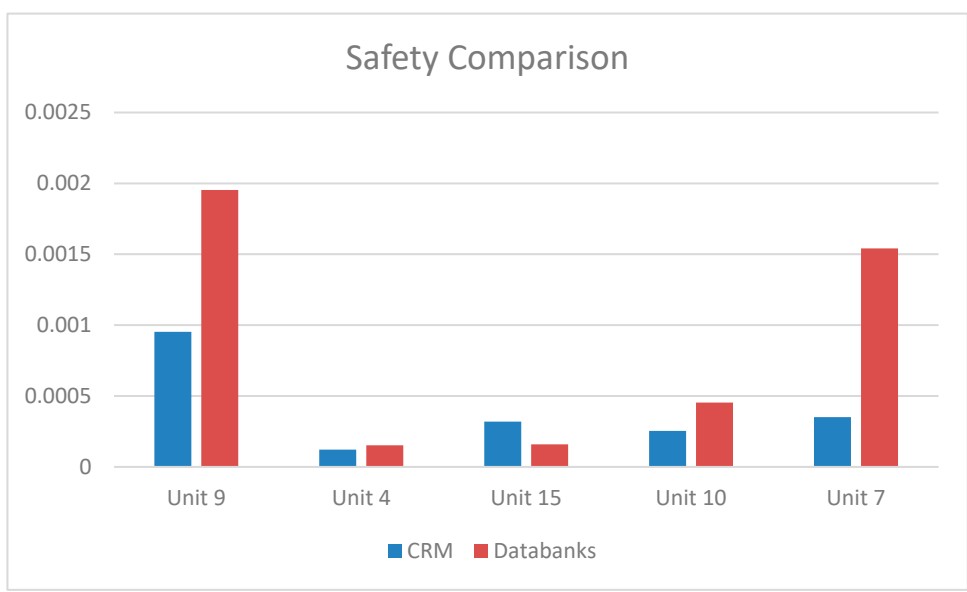

**Figure 8.** Comparison of safety results.

**Table 12.** Safety comparison between CRM and databanks (Mean Absolute Deviation (MAD) and negative technological errors).

| | CRM | Databanks | $\varepsilon$ | $\varepsilon_{\text{tech}}$ (%) | RPN | Databanks | $\varepsilon$ (%) | $\varepsilon_{\text{tech}}$ (%) | Sphynx | Databanks | $\varepsilon$ (%) | $\varepsilon_{\text{tech}}$ (%) |
|---|---|---|---|---|---|---|---|---|---|---|---|---|
| Unit 9 | $9.53 \times 10^{-4}$ | $1.95 \times 10^{-3}$ | $1.00 \times 10^{-3}$ | $-1.00 \times 10^{-3}$ | $1.41 \times 10^{-4}$ | $1.95 \times 10^{-3}$ | $1.81 \times 10^{-3}$ | $-1.81 \times 10^{-3}$ | $1.28 \times 10^{-4}$ | $1.95 \times 10^{-3}$ | $1.82 \times 10^{-3}$ | $-1.82 \times 10^{-3}$ |
| Unit 4 | $1.22 \times 10^{-4}$ | $1.52 \times 10^{-4}$ | $3.00 \times 10^{-5}$ | $-3.00 \times 10^{-5}$ | $2.11 \times 10^{-4}$ | $1.52 \times 10^{-4}$ | $5.93 \times 10^{-5}$ | $5.93 \times 10^{-5}$ | $2.69 \times 10^{-5}$ | $1.52 \times 10^{-4}$ | $1.25 \times 10^{-4}$ | $-1.25 \times 10^{-4}$ |
| Unit 15 | $3.19 \times 10^{-4}$ | $1.59 \times 10^{-4}$ | $1.60 \times 10^{-4}$ | $1.60 \times 10^{-4}$ | $5.22 \times 10^{-5}$ | $1.59 \times 10^{-4}$ | $1.07 \times 10^{-4}$ | $-1.07 \times 10^{-4}$ | $2.83 \times 10^{-5}$ | $1.59 \times 10^{-4}$ | $1.31 \times 10^{-4}$ | $-1.31 \times 10^{-4}$ |
| Unit 10 | $2.54 \times 10^{-4}$ | $4.54 \times 10^{-4}$ | $2.00 \times 10^{-4}$ | $-2.00 \times 10^{-4}$ | $6.26 \times 10^{-5}$ | $4.54 \times 10^{-4}$ | $3.92 \times 10^{-4}$ | $-3.92 \times 10^{-4}$ | $1.02 \times 10^{-4}$ | $4.54 \times 10^{-4}$ | $3.52 \times 10^{-4}$ | $-3.52 \times 10^{-4}$ |
| Unit 7 | $3.51 \times 10^{-4}$ | $1.54 \times 10^{-3}$ | $1.19 \times 10^{-3}$ | $-1.19 \times 10^{-3}$ | $6.52 \times 10^{-5}$ | $1.54 \times 10^{-3}$ | $1.48 \times 10^{-3}$ | $-1.48 \times 10^{-3}$ | $2.73 \times 10^{-5}$ | $1.54 \times 10^{-3}$ | $1.51 \times 10^{-3}$ | $-1.51 \times 10^{-3}$ |
| | | $\text{MAD}_{\text{IFM}}$ | $5.16 \times 10^{-4}$ | $Tot\text{—}\varepsilon_{t \bullet 10chIFM}$ $= -2.42 \times 10^{-3}$ | | $\text{MAD}_{\text{FOO}}$ | $7.69 \times 10^{-4}$ | $Tot\varepsilon_{t \bullet 10chFOO} =$ $-3.79 \times 10^{-3}$ | | $\text{MAD}_{\text{Karmiol}}$ | $7.84 \times 10^{-4}$ | $Tot\varepsilon_{t \bullet 10chKarmiol}$ $= -3.95 \times 10^{-3}$ |

However, the shortcomings of the CRM are:

- Factors $A_2$ and $A_3$ are quali-quantitative values;
- Factor $A_5$ is a qualitative value and is difficult to evaluate without an expert judgement.

These limitations highlight some allocated values greater than those of the databanks (e.g., Unit 15). In this case, the units' performance and hierarchy are not respected. Future research aims to define a quantitative approach for factors $A_2$, $A_3$, and $A_5$.

By comparing the results, the proposed methodology points out results that are more similar to those of databanks, respecting and highlighting hierarchies of performance among units. The reason is simple: The new approach has been structured for series and parallel systems, not only for series ones. This allows important economical savings, since the system's units required less restrictive allocation values.

## 6. Conclusions

In this research, we analyzed a safety allocation issue in a critical infrastructure with many series–parallel units. The conventional safety allocation approaches were developed for critical infrastructures with series configurations, but not for series and parallel ones. The output is an increase of safety allocated to subsystems in series in order to ensure the safety target. In reality, designing and manufacturing a subsystem with an extremely high safety rate would consume a considerable amount of economic resources. The aim of the present paper was to overcome the limitations of the techniques from the literature. The proposed technique was applied in a nuclear infrastructure. By comparing the CRM results with those of conventional methods in terms of MAD and $\varepsilon_{\text{technological}}$, we validated the CRM. The comparison pointed out that CRM provides outputs more similar to those obtained with real data. The new approach points out safety values that are more suitable to databanks and permits a more economical unit design.

The main advantages of the CRM are highlighted below:

- The CRM solves the fundamental problem (parallel configurations) by using new indexes ($A_1$, $A_2$, $A_3$, $A_4$, $A_5$, and $A_6$);
- The CRM results allow the efficient allocation of safety values, meeting customer needs, controlling reasonable support costs, and decreasing manufacturing and maintenance costs [31].
- The comparison with literature is described as follows:
- The MAD of the CRM is smaller than the MAD of the literature methods and is equal to $5.16 \times 10^{-4}$ (failures/year).
- The $\sum(-\varepsilon\text{technological})$ of the CRM is smaller than the $\sum(-\varepsilon\text{technological})$ of the literature methods and is equal to $-2.42 \times 10^{-3}$ (failures/year).

**Author Contributions:** The manuscript was approved by all authors for publication. G.D.B. conceived and designed the study. L.S. collected and analyzed the data. G.D.B. and A.F. wrote the paper. D.F. reviewed and edited the manuscript. All authors have read and agreed to the published version of the manuscript.

**Funding:** This research received no external funding

**Acknowledgments:** We are thankful for the suggestions and efforts of the experts and editors.

**Conflicts of Interest:** The authors declare no conflict of interest.

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
