# Peer review of "Critical Risks Method (CRM): A New Safety Allocation Approach for a Critical Infrastructure"

_sustainability, doi:10.3390/su12124949_

Round 1

Reviewer 1 Report

This is a good paper, and you’ve clearly thought about the important safety implications surrounding critical infrastructures and the measurement of their risks. I particularly like your comparison of differing risk measurement tools.

However, there are some areas that could be used to improve your paper. See below:

Abstract – review language, grammar and formatting. Awkward phrasing throughout. I’d recommend giving the whole paper a good sweep to find grammatical errors, capitalisations (as I could see quite a few while reading).

Unclear sentences: ‘the outcome is an increase of the safety units to ensure the safety target.’ Vague- consider rewording.

Throughout the paper, your say “CRM method” which essentially translates Critical Risk Method method. You should say ‘CRM” without the method afterward, or just write out the whole thing: “Critical Risk Method…”

Page 2: More introduction as to why you introduce the fundamentals of the nuclear system – not clear why you delve into this discussion.

Page 4: what is wrong with qualitative methodology- should explain why this is the case as there are many articles arguing against the positivist approach too. You state over and over again that quantitative methodologies are stronger, but you don’t reference anything to explain why.

Page 3: Were the figures your own? If so, figure 2 is impressive! If not, then you’ll need to cite where it came from.

Line 199: Suggest placing a reference here. An example is Kosovac, A., Davidson, B., & Malano, H. (2019). Are We Objective? A Study into the Effectiveness of Risk Measurement in the Water Industry. Sustainability, 11(5), 1279. doi:10.3390/su11051279 (the analysis finds that there is a lack of objectivity in these matrix type of assessments)

Your outline of the various risk processes are good, and highlights how there are many drawbacks in existing ways of measuring risk.  

Careful of random capitalisations throughout.

Lien 160-161: need citation.

Many of the readers of this journal will not necessarily be risk experts, so it’s a good idea to expand some of the acronyms again (eg. SBD on line 323, and PHA on line 325 for example)

Line 336: I’m not sure whether you’ve explained how you’ve chosen the factors that are included in your analysis- I couldn’t find it at least. But if it isn’t included in your paper, a quick outline would be good here of why you chose criticality, enviro risks, technological, functional, complexity factors

Line 447: This could be a good place to put in any supposed drawbacks of your process. What does the CRM not consider? Is it only a viable alternative for certain systems, and for assessing risk of technological failure? Including this will show that you are not only being critical of other methods, but also of your own (making the paper more robust).

Author Response

Response to Reviewers

Critical Risks Method (CRM): a new safety allocation approach for a critical infrastructure

MANUSCRIPT NUMBER: sustainability-817273

Sustainability – Special Issue: Safety and Security in Critical Infrastructures

Reviewer(s)' Comments to Author:

_________________________________________________________Your Editorial Decision:

Pending major revisions

________________________________________________________________

RECOMMENDATION AND COMMENTS  Review#1

This is a good paper, and you’ve clearly thought about the important safety implications surrounding critical infrastructures and the measurement of their risks. I particularly like your comparison of differing risk measurement tools.

Authors: Thank you to reviewer#1 for giving us the opportunity to submit a revised draft of the manuscript.

We appreciate the time and effort that you and the reviewers have dedicated to provide your valuable feedback on our manuscript ware grateful to the reviewers for their useful suggestions.

We have been able to incorporate changes to reflect most of the suggestions provided by the reviewers and we have highlighted the changes within the manuscript

Here is a point-by-point response to the reviewers’ comments and concerns.

However, there are some areas that could be used to improve your paper. See below:

Abstract – review language, grammar and formatting. Awkward phrasing throughout. I’d recommend giving the whole paper a good sweep to find grammatical errors, capitalisations (as I could see quite a few while reading).

Authors: Language grammar an formatting have been carefully reviewed

Unclear sentences: ‘the outcome is an increase of the safety units to ensure the safety target.’ Vague- consider rewording.

Authors: The above sentence has been modified

Throughout the paper, your say “CRM method” which essentially translates Critical Risk Method method. You should say ‘CRM” without the method afterward, or just write out the whole thing: “Critical Risk Method…”

Authors: The word "method" was deleted after the acronym CRM

Page 2: More introduction as to why you introduce the fundamentals of the nuclear system – not clear why you delve into this discussion.

Authors: Section 2 has been modified

Page 4: what is wrong with qualitative methodology- should explain why this is the case as there are many articles arguing against the positivist approach too. You state over and over again that quantitative methodologies are stronger, but you don’t reference anything to explain why.

Authors: this part havs been better explained at the end of page 4.

Page 3: Were the figures your own? If so, figure 2 is impressive! If not, then you’ll need to cite where it came from.

Authors: The source of Figure 2 has been cited

Line 199: Suggest placing a reference here. An example is Kosovac, A., Davidson, B., & Malano, H. (2019). Are We Objective? A Study into the Effectiveness of Risk Measurement in the Water Industry. Sustainability, 11(5), 1279. doi:10.3390/su11051279 (the analysis finds that there is a lack of objectivity in these matrix type of assessments)

Authors: The suggested reference was mentioned

Your outline of the various risk processes are good, and highlights how there are many drawbacks in existing ways of measuring risk.  

Authors: We are thankful to reviewer 1

Careful of random capitalisations throughout.

Authors: A review has been done

Lien 160-161: need citation.

Authors: A reference has been included

Many of the readers of this journal will not necessarily be risk experts, so it’s a good idea to expand some of the acronyms again (eg. SBD on line 323, and PHA on line 325 for example)

Authors: The acronyms have been expanded

Line 336: I’m not sure whether you’ve explained how you’ve chosen the factors that are included in your analysis- I couldn’t find it at least. But if it isn’t included in your paper, a quick outline would be good here of why you chose criticality, enviro risks, technological, functional, complexity factors

Authors: The factors Ai have been better explained in step 5 of section 2

Line 447: This could be a good place to put in any supposed drawbacks of your process. What does the CRM not consider? Is it only a viable alternative for certain systems, and for assessing risk of technological failure? Including this will show that you are not only being critical of other methods, but also of your own (making the paper more robust).

Authors: The shortcomings of CRM have been described

Reviewer 2 Report

Dear Authors,

first of all let me express my compliments for the interesting subject of your research.

I do not have serious negative considerations about your paper in its structure, results showed and analysis about them. 

I have only some comments and request about the introductory part about the fusion process fenomenology which according to my personal opinion should be rewritten more carefully using more appropriate scientific terms and concepts.

here there are my comments/observations in details: 

abstract

6th line from the beginning: please overcame with overcome. 

2. Nuclear system

page 2: 

lines: 1-2 from the beginning: the sentence: " ... a firmly lively response...." sounds a little bit esoteric. would you be so kind to explain the concept a little bit better?

line 3: the term "iotas" is a little bit unusual in a scientific paper. 

line 5: the molecules are not "electrical impartial" but electrically neutral. 

line 5: electrons DO NOT CIRCLE. please use a more appropriate scientific jargon. 

line 6: Atoms DO NOT HAVE a core but a nucleus. 

line 6: electron charge DOES NOT REMUNERATE PROTON CHARGE. 

line 6: similarly Deuterium has a NUCLEUS and not a CORE.

line 7: helium DOES NOT HAVE a CORE but a NUCLEUS. the Helium nucleus is an ALPHA PARTICLE and not an ALPHA MOLECULE. 

line 8: "Toward the finish of the response the all-out mass is lower than the interfacing components" is very badly written in the sense the terms that have been used DO NOT HAVE scientific meaning. 

line 8: the term used :DISTINCTION is scientifically meaningless in this context. 

line 11: the specification about the common nuclear reactions (which are not the only two listed) is not necessary. So you can simply delete the lines 11-14. 

line 15: the verb WARM is openly appropriate to describe the process to bring the plasma to the temperature regime favourable to the fusion mechanism. 

line 16: the "... extremely high temperatures (around 108 Celsius)" cited here  seem quite easy to reach also in an ordinary kitchen!

line 17: the plasma cannot be "BOUND" like in personal interactions but can be instead CONFINED. 

line 17: therefore, in the scientific jargon researchers talk about magnetic CONFINEMENT and NOT "MAGNETIC RESTRICTION"

line 18: as you have inserted Fig. 2 please mention it.

line 18. I have given a reading to reference [15] mentioned in the text. 

I do not believe that it is the most helpful for a reader that is interested in understand better how a toroidal machine is constructed and how it works. I would suggest to refer to other sources. 

My sincere and personal idea is that this chapter should be rewritten in a better scientific jargon because actually it is not expressed at a scientific standard at the level of a scientific journal. So in order to do that i suggest  for an  exhaustive and scientifically based introduction to the fusion process the reading of the following textbook: 

A.A. Harms, K. F. Schoepf, G. H. Miley, D. R. Kingdon: Principles of Fusion Energy, World Scientific Publishing Co., Singapore, 2002, ISBN 981-02-4335-9

lines (from the top of the page): 7-10. the description seems not to clear and understandable. would you be so kind to rewrite in a clearer way?

in more details: 

line 8: please use HIGHER instead of GREATER. 

line 8: are you sure about the value here reported of (20 mm H20) ?? for the pressure of a cryostat looks to be pretty high. 

lines 8-9: the sentence "in order stay away from the section of barometrical air" is not clear. please write it clearer. what do you mean by barometrical air?

line 9: the sentence: "Humid air freezes and structures hazardous layers of ice" sounds not clear. please rewrite it more understandable.

3. Analysis of Safety Allocation Methods

page 3. 

line 2 from the beginning of the section: 

page 4. 

in eq. (1) S_i(t) misses a *.

line 6 (from the top): please write SUGGESTED instead of SUGGEST. 

ALARP paragraph: 

lines 8-9 (from the beginning of the paragraph): 

the sentence: "This demonstrates by the risk reduction cost (money, time or effort) is grossly disproportionate to the risk reduction gained" is pretty unclear. Please explain it better.

fig.4: on the right hand side some figures are reported. 

I guess that those numbers are the assessed risk values for the different categories of public and workers, which are translated into Risk tolerance criteria as suggested by the UK HSE, right?

line 7 (from the bottom of the paragraph): 

it is more correct to write: operation risks are properly managed according ALARP

page 5. 

line 3 (from the top of the page): what do you mean by gradation? I do not believe that it is the more appropriate word but i am not able to suggest a different one as i do not understand what you are trying to say. 

SPHYNX paragraph 

page 7. 

line 10 (from the bottom of the page): what do you want to say with "it is not realistic the sum of..."?

page 8

line 9 (from the top of the page): established and NOT founded. 

4. CRITICAL RISKS METHOD: 

page 8

line 2 (from the beginning of the section): cooled DOWN and not cooled UP.

page 9

table 4:

please move the caption to the same page of the table and check the style of the caption. 

in the column TOP EVENT why the T.E.3: Minor is followed by (1)? Is it meaning a note or something similar? I could not find this note or remark. 

at the  top of the column SAFETY TARGET (faults/year) it appears (2mission/year)? 2mission is a misprint or it means somerhing? how the figures in this column are evaluated? how long a mission last?  

in this page the table that you are indicating in the caption with  number  6 has to be numbered 5, instead.

page 11

line 6 (from the top of the page): please correct SDB with SBD. 

page 12

what is the numerator in the fraction (23) ? the writing is pretty obscure. 

in the definition of the functional factor H_i in (24) the numerator is different from the denominator in (23)? If yes, why? can you explain it please. 

line 2 from the bottom of the page: after Eq.(1) it is better to write a "." instead of a ":" 

I wish you good luck. 

Author Response

Response to Reviewers

Critical Risks Method (CRM): a new safety allocation approach for a critical infrastructure

MANUSCRIPT NUMBER: sustainability-817273

Sustainability – Special Issue: Safety and Security in Critical Infrastructures

Reviewer(s)' Comments to Author:

_________________________________________________________Your Editorial Decision:

Pending major revisions

________________________________________________________________

RECOMMENDATION AND COMMENTS  Review#2

Dear Authors, first of all let me express my compliments for the interesting subject of your research. I do not have serious negative considerations about your paper in its structure, results showed and analysis about them. 

I have only some comments and request about the introductory part about the fusion process phenomenology which according to my personal opinion should be rewritten more carefully using more appropriate scientific terms and concepts. Here there are my comments/observations in details: 

Authors:

Thank you to reviewer#2 for giving us the opportunity to submit a revised draft of the manuscript.

We appreciate the time and effort that you and the reviewers have dedicated to provide your valuable feedback on our manuscript ware grateful to the reviewers for their useful suggestions.

We have been able to incorporate changes to reflect most of the suggestions provided by the reviewers and we have highlighted the changes within the manuscript

Here is a point-by-point response to the reviewers’ comments and concerns.

Abstract: 6th line from the beginning: please overcame with overcome

Authors: The correction has been done

  1. Nuclear system - page 2: 

lines: 1-2 from the beginning: the sentence: " ... a firmly lively response...." sounds a little bit esoteric. would you be so kind to explain the concept a little bit better?

Authors: The sentence has been modified to better explain the concept

line 3: the term "iotas" is a little bit unusual in a scientific paper. 

Authors: The term has been changed

line 5: the molecules are not "electrical impartial" but electrically neutral. 

Authors: The term has been changed

line 5: electrons DO NOT CIRCLE. please use a more appropriate scientific jargon. 

Authors: The term has been changed

line 6: Atoms DO NOT HAVE a core but a nucleus. 

Authors: The term has been changed

line 6: electron charge DOES NOT REMUNERATE PROTON CHARGE. 

Authors: The sentence has been changed

line 6: similarly Deuterium has a NUCLEUS and not a CORE.

Authors: The term has been changed

line 7: helium DOES NOT HAVE a CORE but a NUCLEUS. the Helium nucleus is an ALPHA PARTICLE and not an ALPHA MOLECULE. 

Authors: The terms have been changed

line 8: "Toward the finish of the response the all-out mass is lower than the interfacing components" is very badly written in the sense the terms that have been used DO NOT HAVE scientific meaning. 

Authors: The sentence has been changed

line 8: the term used :DISTINCTION is scientifically meaningless in this context. 

Authors: The term has been changed

line 11: the specification about the common nuclear reactions (which are not the only two listed) is not necessary. So you can simply delete the lines 11-14. 

Authors: Lines 11-14 have been deleted

line 15: the verb WARM is openly appropriate to describe the process to bring the plasma to the temperature regime favourable to the fusion mechanism. 

Authors: The term has been changed

line 16: the "... extremely high temperatures (around 108 Celsius)" cited here seem quite easy to reach also in an ordinary kitchen!

Authors: the temperature is 108 Celsius

line 17: the plasma cannot be "BOUND" like in personal interactions but can be instead CONFINED. 

Authors: The term has been changed

line 17: therefore, in the scientific jargon researchers talk about magnetic CONFINEMENT and NOT "MAGNETIC RESTRICTION"

Authors: The term has been changed

line 18: as you have inserted Fig. 2 please mention it.

Authors: The source of Figure 2 has been cited

line 18. I have given a reading to reference [15] mentioned in the text. I do not believe that it is the most helpful for a reader that is interested in understand better how a toroidal machine is constructed and how it works. I would suggest to refer to other sources. My sincere and personal idea is that this chapter should be rewritten in a better scientific jargon because actually it is not expressed at a scientific standard at the level of a scientific journal. So in order to do that i suggest  for an  exhaustive and scientifically based introduction to the fusion process the reading of the following textbook: A.A. Harms, K. F. Schoepf, G. H. Miley, D. R. Kingdon: Principles of Fusion Energy, World Scientific Publishing Co., Singapore, 2002, ISBN 981-02-4335-9

Authors: The reference has been changed and the section 2 has been rewritten in a better scientific jargon

lines (from the top of the page): 7-10. the description seems not to clear and understandable. would you be so kind to rewrite in a clearer way?

Authors: The sentence has been rewritten

in more details: line 8: please use HIGHER instead of GREATER. 

Authors: The term has been changed

line 8: are you sure about the value here reported of (20 mm H20) ?? for the pressure of a cryostat looks to be pretty high. 

Authors: The value has been corrected

lines 8-9: the sentence "in order stay away from the section of barometrical air" is not clear. please write it clearer. what do you mean by barometrical air?

Authors: The sentence has been rewritten

line 9: the sentence: "Humid air freezes and structures hazardous layers of ice" sounds not clear. please rewrite it more understandable.

Authors: The sentence has been rewritten

  1. Analysis of Safety Allocation Methods

page 4. in eq. (1) S_i(t) misses a *.

Authors: This has now been done

line 6 (from the top): please write SUGGESTED instead of SUGGEST. 

Authors: This has now been done

ALARP paragraph: lines 8-9 (from the beginning of the paragraph): the sentence: "This demonstrates by the risk reduction cost (money, time or effort) is grossly disproportionate to the risk reduction gained" is pretty unclear. Please explain it better.

Authors: The sentence has been better explained

fig.4: on the right hand side some figures are reported. I guess that those numbers are the assessed risk values for the different categories of public and workers, which are translated into Risk tolerance criteria as suggested by the UK HSE, right?

Authors: Figure 4 has been changed

line 7 (from the bottom of the paragraph): it is more correct to write: operation risks are properly managed according ALARP

Authors: The sentence has been corrected

page 5. line 3 (from the top of the page): what do you mean by gradation? I do not believe that it is the more appropriate word but i am not able to suggest a different one as i do not understand what you are trying to say. 

Authors: The word “gradation” is changed in “ranking”.

SPHYNX paragraph 

page 7. line 10 (from the bottom of the page): what do you want to say with "it is not realistic the sum of..."?

Authors: The sentence has been changed

page 8 line 9 (from the top of the page): established and NOT founded. 

Authors: This has now been done

  1. CRITICAL RISKS METHOD: 

page 8 line 2 (from the beginning of the section): cooled DOWN and not cooled UP.

Authors: This has now been done

page 9 table 4: please move the caption to the same page of the table and check the style of the caption. 

Authors: This has now been done

in the column TOP EVENT why the T.E.3: Minor is followed by (1)? Is it meaning a note or something similar? I could not find this note or remark. 

Authors: It has been deleted. It is a misprint.

at the  top of the column SAFETY TARGET (faults/year) it appears (2mission/year)? 2mission is a misprint or it means somerhing? how the figures in this column are evaluated? how long a mission last?  

Authors: It has been deleted. It is a misprint.

in this page the table that you are indicating in the caption with  number  6 has to be numbered 5, instead.

Authors: This has now been done

page 11 line 6 (from the top of the page): please correct SDB with SBD. 

Authors: This has now been done

page 12 what is the numerator in the fraction (23)? the writing is pretty obscure. 

Authors: The numerator in fraction (23)

in the definition of the functional factor H_i in (24) the numerator is different from the denominator in (23)? If yes, why? can you explain it please. 

Authors: The numerator in (24) is the same of denominator in (23). The numerator Hi has been better explained.

line 2 from the bottom of the page: after Eq.(1) it is better to write a "." instead of a ":" 

Authors: This has now been done

Round 2

Reviewer 2 Report

Dear Authors, 

i have carefully read the manuscript and i feel satisfied with the improvements you have made.